# Adsorption of Peptides onto Carbon Nanotubes Grafted with Poly(ethylene Oxide) Chains: A Molecular Dynamics Simulation Study

**DOI:** 10.3390/nano12213795

**Published:** 2022-10-27

**Authors:** Zuzana Benková, Peter Čakánek, Maria Natália D. S. Cordeiro

**Affiliations:** 1Polymer Institute, Slovak Academy of Sciences, Dúbravská Cesta 9, 845 41 Bratislava, Slovakia; 2LAQV@REQUIMTE, Department of Chemistry and Biochemistry, University of Porto, Rua do Campo Alegre s/n, 4169-007 Porto, Portugal

**Keywords:** carbon nanotubes, peptide adsorption, antifouling agents, poly(ethylene oxide), polyglycine, polyserine, polyvaline, secondary structure, cylindrical density distributions, hydrogen bonds

## Abstract

Carbon nanotubes (CNTs) display exceptional properties that predispose them to wide use in technological or biomedical applications. To remove the toxicity of CNTs and to protect them against undesired protein adsorption, coverage of the CNT sidewall with poly(ethylene oxide) (PEO) is often considered. However, controversial results on the antifouling effectiveness of PEO layers have been reported so far. In this work, the interactions of pristine CNT and CNT covered with the PEO chains at different grafting densities with polyglycine, polyserine, and polyvaline are studied using molecular dynamics simulations in vacuum, water, and saline environments. The peptides are adsorbed on CNT in all investigated systems; however, the adsorption strength is reduced in aqueous environments. Save for one case, addition of NaCl at a physiological concentration to water does not appreciably influence the adsorption and structure of the peptides or the grafted PEO layer. It turns out that the flexibility of the peptide backbone allows the peptide to adopt more asymmetric conformations which may be inserted deeper into the grafted PEO layer. Water molecules disrupt the internal hydrogen bonds in the peptides, as well as the hydrogen bonds formed between the peptides and the PEO chains.

## 1. Introduction

Since their discovery in 1991 [1], carbon nanotubes (CNTs) have achieved tremendous attention and have been utilized in abundant applications. This carbon allotrope, which may be viewed as a rolled-up version of the graphene sheet, earned great interest due to its unique properties such as high tunable electrical [2] and thermal [3] conductivity, mechanical resistance [4], low density, and useful optical properties [5,6]. Therefore, CNTs are widely employed in industrial applications as components in electronics, energy-storage devices, solar cells, sensors, or as fillers in polymeric composites in mechanical applications [7]. CNTs have also attracted a great interest in the field of biomedicine. CNT-based nanoplatforms were assumed as various biological molecular cargoes into cells, including drugs [8], plasmid DNA [9], peptides, proteins [10], and small interfering RNA [11]. They have also found immense potential in biological imaging techniques [12], tissue engineering [13], and cancer therapy [14]. The main disadvantage of pristine CNTs is their hydrophobicity. It brings about intrinsic insolubility of CNTs in water and in many other solvents and biological media and leads to the aggregation of CNTs, which makes their manipulation more difficult, as well as to toxicity problems [15,16]. Appropriate surface functionalization of CNTs improves dispersity of CNTs and thus overcomes these limitations [17]. The functionalization can be realized through: (1) covalent attachment of suitable molecules (chemical adsorption), (2) non-covalent interactions of molecules with the sidewall of CNTs (physical adsorption), or (3) encapsulation of molecules in the inner cavity of CNTs. In addition to the solubility enhancement, coating of CNTs in biomedical applications must prevent the undesired adsorption of plasma proteins and should prolong the blood circulation time. The protective barrier of CNTs against the formation of protein corona at the CNT interface can be designed with the help of several amphiphiles such as lipids, surfactants, and hydrophilic polymers.

Poly(ethylene oxide) (PEO) belongs to the most frequently utilized antifouling agents, since it is perfectly soluble in water, biocompatible, nontoxic, and is not supposed to induce immune response. A large number of experimental and theoretical studies has been devoted to interactions of PEO layers grafted on planar surfaces and proteins. Concerning the antifouling properties of PEO, the conclusions of these studies are often conflicting. The coating efficiency of PEO is usually attributed to its compatibility with water and biological inertness. The developed theories are based on three types of interactions between the grafted PEO layer and a protein, which could be responsible for antifouling behavior of grafted PEO modified surfaces.

The early theory has considered the interplay of steric repulsion, hydrophobic attractions, and van der Waals attractions between a spherical protein containing a hydrophobic patch and PEO grafted chains adopting the brush conformation in water [18,19]. The authors pointed out that the optimal protein resistance of PEO grafted layer is achieved at high grafting density and long PEO chains; the grafting density was found to be a more relevant factor than the chain length [18,19]. These conclusions have been supported by a number of experimental works [20,21,22,23], but experimental works exist which demonstrated a high antifouling efficiency of short PEO chains grafted on planar solid surfaces [24,25,26,27].

The single-chain mean-field theory has been adopted to study the interactions of lysozyme with PEO modified surfaces displaying various affinities to PEO chains and attractive interactions with lysozyme at contact [28].

According to the shape and size of a protein interacting with grafted PEO chains as well as the specific and non-specific interactions between the protein and the PEO grafted surface, three theoretical models of adsorption are classified, namely, primary, secondary, and ternary adsorptions [29,30,31]. Primary adsorption, typical of small proteins, occurs at the surfaces, secondary adsorption, characteristic of large rod-like proteins, occurs at the edge of the grafted brushes, and ternary adsorption is observed when a globular protein penetrates the brush.

The effect of the grafting density, length, and distal chemistry of PEO chains chemisorbed to gold substrate on antifouling properties have been investigated experimentally using null ellipsometry [32,33]. The distal chemistry plays a role at higher grafting density when the PEO chains are extended away from the supporting surface and their terminal substituents are exposed to water medium. The hydrophilic hydroxyl groups promote the protein repulsion as they compensate the dehydration of the grafted layer at higher grafting densities.

The experimental investigation of self-assembled monolayers (SAMs) of alkanethiols terminated by short PEO chains onto the gold and silver substrates has revealed that SAMs on gold repels fibrinogen, whereas SAMs on silver attracts fibrinogen [34]. These observations have been explained using grand canonical Monte Carlo method [35,36]. 

All aforementioned analyses and findings relate to planar supporting surfaces. The situation is different for curved surfaces such as CNTs. The brush conformation of chains grafted onto the curved surface requires more densely distributed grafting points when compared with the planar substrates. In the case of attractive interactions between the grafted chains and the carbon nanotube at low grafting densities, the grafted chains can be wrapped around the carbon nanotube or aligned with the axis of CNT [37,38,39,40].

The experimental studies of CNT chemically or physically covered with PEO chains have reported accumulation of proteins around PEO modified CNTs independently of the protein molar mass, total hydrophobicity, isoelectric point, number of aromatic residues, and PEO conformation [41]. Other experimental findings have demonstrated only partly reduced in vivo toxicity and slightly improved biocompatibility of CNTs upon their functionalization with PEO chains [42,43]. On the contrary, earlier experimental studies have not found out any evidence of toxicity of PEO functionalized CNTs over 4 months [44] and also an excellent resistance of CNTs to nonspecific protein binding has been gained through a strong physical adsorption of Tween 20 (surfactant comprising a linear aliphatic chain and three PEO branches) and Pluronics P103 (triblock copolymer with poly(propylene oxide), PEO_20_-PPO_52_-PEO_20_) [45]. Coverage of CNTs with phospholipid-PEO chains has imparted in vivo enhanced plasma protein resistance to CNTs [46]. Noncovalent functionalization of CNTs by PEO chains has remarkably reduced the adsorption of serum proteins such as albumin, preambulin, transferrin, and immunoglobulin [47]. PEO modified CNTs have also been suggested as biocompatible and effective nanocarriers for intra-articular delivery of agents for chondrocytes [48]. Functionalization of CNTs with PEO chains has improved binding with drugs such as Doxorubicin through the strong π–π stacking interactions with the aromatic moieties [49,50]. Enhanced drug release rate of Doxorubicin and internalization into cells has been observed for CNTs difunctionalized by poly(ethylene oxide) and poly(ethylene imine) [51]. Pegylated CNTs have also appeared as efficient carriers for Paclitaxel without causing obvious toxic effects [52,53]. Improved biocompatibility of CNTs non-covalently covered by phospholipid-PEO polymers, which are used for the delivery of Cyclosporin A through its covalent bonding with PEO, has been also corroborated [54]. PEO modified CNTs in conjugation with the integrin *α_v_β_3_* monoclonal antibody have been considered as transporters for the detection of cancer cells [55]. The phospholipid-PEO physically grafted CNTs have been employed to deliver short interfering RNA to mammalian cells [56] or as near-infrared agents for selective cancer-cell destruction [57]. Pegylated CNTs have been also introduced as potential hem-compatible and nontoxic curcumin transporters in experiments with mice fibroblast cells [58], and the increased adsorption capacity and prolonged released property of curcumin transported by pegylated CNTs in vitro in different human body fluids have been confirmed in another experimental study [59].

Although currently, there are plenty of experimental techniques enabling us to understand the adsorption of proteins onto PEO modified CNTs, the resolution dimension limit is about 1 nm, which hinders structural and dynamic analyses at molecular and atomic levels. Investigation of phenomena and processes at these length scales is possible using atomistic or coarse-grained molecular simulations. Specifically, molecular dynamics (MD) simulations have become a powerful tool to understand the experimental findings by providing insights into these systems at atomistic level. Up-to-date, MD studies have been preferentially focused on PEO modified CNTs in aqueous medium. Essentially, such studies tackled the adsorption of PEO chains on CNTs in water [60,61,62], the dispersion of PEO covered CNTs [60,63,64], and the influence of PEO grafting density and size on the conformation of PEO layer covalently or noncovalently grafted on CNTs [65,66]. Later on, larger systems incorporating PEO modified CNTs interacting with some other molecules such as plasma proteins or drug molecules have also been studied employing molecular simulations. In comparison to computational studies of the systems composed of the PEO-modified CNTs as carriers of anticancer drugs [67,68,69,70,71,72], little attention has been paid to the PEO-modified CNTs interacting with proteins or peptides. The effect of protein shape and PEO length and grafting density, as well as the type of PEO-CNT grafting on the adsorption of fibrinogen and serum albumin onto PEO-modified CNTs, have been studied using coarse-grained molecular dynamics simulations [73]. The formation of a protein corona around pegylated CNTs have mitigated the observed inhibition of CP3A4 enzymatic activity [74].

The interactions of a protein with PEO grafted surfaces represents a complex problem which encompasses many factors such as the conformation of the grafted polymer chains, the alteration of this conformation upon the protein approach, the protein shape, size, and orientation with respect to the PEO layer, the modification of the protein conformation and shape during its interaction with the grafted layer, the hydration of grafted layer, the changes in hydration layer during the protein approach, hydrophobicity/hydrophilicity and inhomogeneity of a protein, and specific interactions such as hydrogen bonding or π–π stacking interactions. Due to this large variety of independent or coupled factors, the mechanism of PEO based protein resistance has not yet been satisfactorily elucidated.

In this study, atomistic molecular dynamics (MD) simulations have been performed to examine the interactions of CNT irreversibly grafted with PEO chains at different grafting densities with peptides containing neutral substituents, in dry and aqueous conditions. The effect of PEO coverage of CNT, hydration of the PEO layer, and the presence of NaCl on peptide adsorption is of interest.

## 2. Methodology

The zig-zag configuration of an uncapped CNT with the chirality index (10, 0), the tube diameter of 0.8 nm and the tube length of 4.1 nm was selected as a sufficiently curved supporting substrate for PEO chains. The total area of the CNT sidewall was 10.3 nm^2^. The PEO chains comprising *N* = 18 monomers were terminated by methoxy groups and irreversible anchored to the CNT by one end. The grafting densities of PEO chains, expressed as the number of chains attached to a unit area, were *σ* = 0.388, 0.776, and 1.553 nm^−2^. These grafting densities were achieved by attaching 4, 8, and 16 chains onto the CNT. Periodic boundary conditions were applied in all three dimensions. The size of the cubic simulation box was 8–9 nm in peptide-free systems and 11–12 nm in peptide-containing systems. In the case of pristine CNTs, the size of the cubic simulation box was 6–6.5 nm. In the initial conformation, the stretched PEO helices were radially orientated and evenly distributed over the CNT sidewall as shown in Figure 1. Three peptides built up of 24 identical amino acid residues of various hydrophobicity were considered, namely: polyvaline as a hydrophobic representative with an aliphatic side chain, polyglycine as a flexible hydrophobic representative without side chains, and polyserine as a hydrophilic representative with neutral side chains where OH group may serve as a protein donor or acceptor in hydrogen bonds. The hydropathy indices of valine, glycine, and serine are 4.2, −0.4, and −0.8, respectively [75]. Although serine has a similar hydropathy index to glycine, serine contains a hydroxyl group that is responsible for specific interactions with water and the PEO chains. The terminal amino and carboxyl groups were assumed in their protonated and dissociated forms, respectively. Atoms of the CNT were kept fixed at their equilibrium positions during the MD simulations and participated only in nonbonded interactions with other system constituents, i.e., PEO chains, peptides, and water molecules. The CNT axis was aligned with the *z* Cartesian coordinate. The interactions of peptides with PEO covered CNTs were simulated in vacuum, water, and saline (physiological solution of NaCl, 9.0 g·dm^−3^) in order to unveil the effect of water and presence of ions. The interactions of peptides with pristine CNTs in water and saline were simulated as well in order to address the effect of PEO coverage on the structure of adsorbed peptides. The number of water molecules varied depending on the simulation box size from 7010 (simulation box size 6 nm) to 55,966 (simulation box size 12 nm). In the initial geometry, the peptides were orientated to achieve the maximal close contacts between the peptides and the PEO chain or CNT sidewall in the case of pristine CNT. One simulation run was performed for each system.

All MD simulations were carried out using the GROMACS 5.1 package [76]. The all-atom intra- and intermolecular interactions were described by the OPLSAA force field [77,78,79]. The sp^2^ hybridized carbon atoms of CNT were treated as uncharged and nonpolarizable, thus only van der Waals nonbonded interactions of CNT with other atoms came into consideration. Water molecules were represented by the TIP3P model [80,81].

The systems were simulated in the *NVT* ensemble at temperature *T* = 298 K. The temperature was controlled by means of a canonical velocity-rescaling thermostat [82] with a relaxation time of 0.1 ps. First, the CNT grafted with PEO chains at all grafting densities in water and peptides in water were separately pre-equilibrated for 10 ns in the *NVT* ensemble at temperature *T* = 298 K. In such simulations, the time step used was 1 fs and the time interval of conformation sampling was 10 ps. The long-range electrostatic interactions were computed with the particle-mesh Ewald method [83,84] with the real space cut-off set to 1.4 nm. The Lennard-Jones potential used for the calculations of the short-range van der Waals interactions was smoothly shifted to zero between 1.0 and 1.4 nm. To find out the adequate equilibration period for PEO grafted CNTs, two initial conformations of PEO-CNT system, where PEO chains were not immobilized by their ends on the CNT, were simulated in water. In the first system, the helical axes of the PEO chains were parallelly aligned to the CNT axis, whereas in the second system the helical axes of the PEO chains were oriented radially, i.e., perpendicularly to the CNTs axis as shown in Figure 1. After the physical adsorption in both conformations, the time when both systems achieved the same cylindrical density distributions and identical (within standard deviations) structural parameters of PEO chains was used for estimation of the equilibration period. This time was less than 10 ns for all three grafting densities considered in this study. The CNT was able to accommodate up to 16 PEO chains. After removal of the water molecules from both systems, the peptide was placed in close vicinity to the PEO covered CNT so that the distance between the nearest PEO and peptide atoms was about 0.3 nm. This composite system was used as a starting conformation of simulations in all three environments, i.e., in vacuum, water, and saline. The size of the cubic simulation box was 11–12 nm in all investigated systems. The equilibration period of 10 ns was followed by a production run of 30 ns, using the same time step (1 fs).

## 3. Results and Discussion

### 3.1. Density Distributions

The cylindrical density distributions of the grafted PEO chains and the peptides around the CNT axis are shown in Figure 2. These density distributions are expressed as the number of atoms in the volume elements of cylindrical shells of thickness 0.1 nm and of length 4.1 nm concentrically distributed around the CNT axis. Only nonhydrogen atoms are considered in the case of PEO chains whereas all atoms are considered in the case of peptides. The distance of the CNT sidewall from the CNT axis is also marked on the *x* coordinate. With the exception of polyglycine and polyserine interacting with the CNT coated by the PEO chains at *σ* = 0.388 nm^−2^ in dry conditions, the cylindrical density distributions of the grafted PEO chains are not affected by the presence of any peptide as revealed by a comparison of the cylindrical density distributions (Figure 2) with the cylindrical density distributions of the PEO chains in the absence of peptides (Appendix A in the Appendix A). At the lowest grafting density, *σ* = 0.388 nm^−2^ in a vacuum, the cylindrical density distribution of PEO chains becomes slightly more extended from the CNT sidewall in the presence of polyglycine and polyserine instead of being compressed. In a vacuum, at the two higher grafting densities, the second peak is formed in the cylindrical density distributions of the PEO chains which preserves only as a shoulder in water and saline. These constant density distributions of the PEO chains in all investigated environments for a given grafting density, regardless of the different depths of peptide insertion, suggest that the peptides adhere to the grafted PEO chains without compressing the PEO layer. As can be seen in Figure 2, the peptides are inserted in the layer of grafted PEO chains at higher grafting density also in water and saline, which is not reflected on the cylindrical density distributions of PEO chains. In water and saline, some atoms of the peptides can fill the space at the expense of the expelled water molecules as shown in Appendix A. The entropy gain related to this release of water molecules from the PEO layer may promote the adsorption of peptides in water and in saline. Inspection of Appendix A reveals that at *σ* = 1.553 nm^−2^, there is only a marginal reduction in the interfacial water density resulting from the peptides adsorption, especially in the first hydration shell. At this grafting density, fewer water molecules are incorporated in the PEO layer; therefore, fewer water molecules are expelled upon peptide adsorption. A reduction in the cylindrical density distribution of water in the vicinity of pristine CNT induced by the grafted PEO layer at *σ* = 0.388 and 1.553 nm^−2^ is shown in Appendix A. However, the intensity of the water density in the hydration layers drops, the structuring of the interfacial water layer induced by CNT is retained after functionalization of CNT with the PEO chains as well as after the adsorption of the peptides. While the coating of CNT with PEO chains notably affects the intensity of the interfacial water density distribution, the adsorption of peptides brings about only modest reduction in the intensity of the interfacial water density distribution. This is evident from Appendix A, which illustrates the cross-sectional view of the systems with water molecules arranged around CNT covered by the PEO chains (not shown for clarity) grafted at different grafting densities in the absence of any peptide and in the presence of polyglycine. For comparison, the cross-sectional view of the interfacial water molecules around pristine CNT with adsorbed polyglycine is also incorporated. From Appendix A, it is clear that the presence of polyglycine does not introduce appreciable changes.

Although the investigated peptides have different affinity for water, a notable influence of the medium on the peptide adsorption in the case of polyserine and polyvaline is observable only at the lowest grafting density. The presence of NaCl at physiological concentration has only a small effect on the adsorption of all investigated peptides with the exception of polyserine, which in saline, penetrates deeper into the layer of PEO chains grafted at *σ* = 0.388 nm^−2^. In fact, at this grafting density, polyserine reaches the CNT sidewall as in the case of pristine CNT. Analysis of the radial distribution functions (RDFs) provides an explanation for this behavior. In addition to the three types of expected protein-salt electrostatic interactions including O_(carboxyl)_–Na^+^, N-Na^+^, and H_(amide)_–Cl^−^ pairs in the investigated peptides, two other types of electrostatic interactions exist in polyserine, namely, O_(hydroxyl)_–Na^+^ and H_(hydroxyl)_–Cl^−^. The radial distribution functions for atoms of the three peptides interacting with PEO chains grafted at *σ* = 0.388 nm^−2^ are presented in Figure 3. The absence of a peak in the RDF of N-Na^+^ pairs indicates that the sodium cation preferentially interacts with oxygen atoms of peptides. Only in the case of polyglycine interacting with CNT covered by the PEO chains at *σ* =1.553 nm^−2^, where substantially enhanced density of sodium cations at the CNT interface is observed when compared with the analogous system containing polyserine or polyvaline (Appendix A), intense peaks due to N–Na^+^ and O–Na^+^ pairs appear in the RDF (see Appendix A). The sodium cations may form bridges between the oxygen atoms of the PEO chains and peptides. There are two types of interactions between oxygen and sodium cation in the case of polyserine instead of only one type in the case of polyglycine and polyvaline. Moreover, the peak of RDF of O_(carboxyl)_–Na^+^ pairs is of higher intensity than that of polyvaline. The cylindrical density distributions of the sodium cations and chloride anions are presented in Figure 4a. There is an evident peak in the cylindrical density distribution of the sodium cations whose position almost coincides with the peaks in the cylindrical density distributions of the oxygen atoms in the PEO chains and polyserine (Figure 4b). This implies that a sodium cation may act as a bridge between an oxygen atom of the PEO chains and an oxygen atom of polyserine. The intensity of this peak drops with increases in the grafting density.

In water as well as in saline, polyglycine is inserted deeper into the PEO layer than the more hydrophobic polyvaline at *σ* = 0.388 nm^−2^. This can be attributed to the smaller size of polyglycine and its larger flexibility that allows polyglycine to adopt its conformation for optimal insertion into the PEO layer. In fact, at *σ* = 0.388 nm^−2^ and to a lesser extent at *σ* = 0.776 nm^−2^ and 1.553 nm^−2^, polyglycine penetrates deeper to the CNT sidewall in water and saline than in vacuum. A comparison of polyglycine adsorbed on pristine CNT with polyglycine adsorbed on CNT grafted with the PEO chains at *σ* = 0.388 nm^−2^ in water and saline (Figure 2a) unveils that also in the latter case, the polyglycine comes into direct contact with the CNT sidewall, even closer than in a vacuum. It is caused by the extension of PEO chains from CNT into water or saline which creates a space for relatively small polyglycine to penetrate through the PEO layer to the CNT sidewall and to maximize the number of favorable contacts of polyglycine with the CNT sidewall and PEO atoms.

Worth mentioning is also the closer approach of polyvaline to the CNT sidewall than polyserine and polyglycine in a vacuum in spite of its bulkier size. This might be ascribed to more efficient insertion of hydrophobic polyvaline into the PEO layer due to van der Waals interactions of hydrophobic polyvaline with PEO coated CNT. At this point, it should be mentioned that, besides the methyl substituents of polyvaline which seem to be preferentially attracted by the CNT sidewall regardless of the medium and the presence or absence of the PEO chains, there is no preference of either PEO modified or pristine CNT to attract some specific atom of polyglycine or polyserine.

### 3.2. Hydrogen Bonds and Secondary Structure of Peptides

The number of intramolecular hydrogen bonds formed within the investigated peptides and the number of hydrogen bonds formed between the peptides and water molecules are shown in Figure 5 in vacuum, water, and saline. The pure geometric distance-angle criterion was adopted for definition of hydrogen bonds, according to which a hydrogen bond exists if the acceptor-donor distance is not more than 0.35 nm and the maximal plausible deviation of the acceptor-donor-hydrogen angle from linearity is 30°. Polyserine with its hydroxyl groups is expected to participate in formation of more hydrogen bonds than polyglycine and polyvaline. The formation of hydrogen bonds in polyvaline is hindered by two methyl substituents present in the amino acid residue. It is obvious that in water as well as in saline, the water molecules disrupt the intramolecular hydrogen bonds formed within the peptides at all grafting densities which is more striking for polyvaline (Figure 5a–c). The number of hydrogen bonds between the peptides and the water molecules in the absence of PEO chains, which is the case of the free peptides in water or the peptides interacting with pristine CNT in water or in saline, tends to be slightly higher than in the presence of the PEO chains and either decreases with increasing grafting density in the case of polyglycine in water and saline and polyserine in saline or remains stable in the case of polyserine in water or polyvaline in water and saline (Figure 5d–f). The competition of hydrogen bonds formed between the peptides and water molecules with hydrogen bonds formed between the peptides and the PEO chains, especially in polyserine and polyvaline, can also be seen in Appendix A, which shows the number of hydrogen bonds between the peptides and the PEO chains. Polyglycine displays the highest tendency to form hydrogen bonds with the PEO chains in water and in saline. This might be explained by its more efficient insertion into the grafted PEO layer. In the case of polyvaline, there is also the steric hindrance by two methyl substituents in each amino acid residue that prevents the formation of hydrogen bonds between the PEO oxygen atoms and the amide hydrogen atoms. Interestingly, the enhanced number of hydrogen bonds created between polyvaline and the PEO chains at *σ* = 0.776 nm^−2^ in a vacuum (see Appendix A) correlates with the diminished number of intramolecular hydrogen bonds of polyvaline at this grafting density (Figure 5c). As expected, polyserine with the hydroxyl groups in its side chains creates the most intramolecular hydrogen bonds and hydrogen bonds with water molecules. On the other hand, the presence of the peptides does not seem to be capable of breaking hydrogen bonds between the PEO chains and the water molecules (Appendix A).

The secondary structure of proteins and peptides is maintained through the intramolecular hydrogen bonds, electrostatic, and van der Waals interactions. Thus, it is of interest to study the evolution of the secondary structure of the investigated peptides during their interactions with and adsorption on pristine as well as PEO-covered CNTs in various environments. The secondary structure has been analyzed using the STRIDE algorithm [85], which combines hydrogen bond energy data with statistically derived backbone torsion angle data. A weighted product of hydrogen bond energy and torsion angle probabilities for α-helix and β-sheet determines the initiation and termination of secondary structure units based on empirically optimized recognition thresholds.

Conformational preference of individual amino acids in globular proteins to specific secondary structure elements has been reported based on the X-ray crystallographic data [86]. According to this study, valine with its two methyl groups is prone to adopt β-sheets, whereas serine with its polar substituent and glycine with no chain substituent tend to constitute reverse turns. On the other side, according to the DFT study of polyglycine in a vacuum, α-helix has been pointed out as the most stable element of secondary structure among the considered α-helix, 3_10_-helix, β-sheet, and γ-turn [87]. However, inspection of the basis set superposition error has revealed that, for a medium-sized basis set, this error is of the same order of magnitude as the relative energy difference of secondary structure elements [88]. This finding makes the ab initio gas-phase predictions of stability of homopeptides doubtful. Discontinuous molecular dynamics simulations of polyglycine assuming solvent effects implicitly have rendered β-sheets as more stable structural elements than helical fragments [89]. According to the presented MD simulations, the secondary structure of the free peptides in water consist of turns and coils, which is enriched by two β-sheet fragments in polyglycine.

In a vacuum, where the number of intramolecular hydrogen bonds within the peptides is not impaired by water molecules, the secondary structure of all three peptides remains stable during their adsorption on the PEO coated CNTs. Apart from the turns and coiled fragments, which are salient features of all three investigated peptides in all three environments, polyglycine possesses two short fragments of β-sheet which are extended upon the interactions with more densely grafted PEO chains in a vacuum. In water and saline, the β-sheets in polyglycine are either suppressed or transformed into isolated bridges in the presence of PEO modified CNT. This is in contrast with the observed stabilization of the β-sheets during the adsorption of proteins onto planar graphene substrate [90,91]. When polyglycine interacts with pristine CNT, the β-sheets are stabilized in saline and replaced by isolated bridges in water.

At *σ* = 0.388 nm^−2^ in a vacuum, polyserine exhibits only turns and coils, whereas at the two higher grafting densities, one 3_10_-helix arises. The 3_10_-Helix is also observed in polyserine interacting with PEO chains grafted at the two higher grafting densities in water and saline and at the lowest grafting density in saline, moreover, the fragments of isolated bridges are generated in water at all grafting densities. Generation of the 3_10_-helix might be explained by the more compact conformation of polyserine trapped among the grafted PEO chains. The adsorption of polyserine onto pristine CNT stabilizes the 3_10_-helix and isolated bridges in saline; however, these secondary structure elements periodically appear and disappear in water. Stabilization of helical structures encountered in oligoserine composed of 13 amino acid residues in water when compared with a vacuum has been reported [92]. The authors have adopted the MD method and ascribed this stabilization to breakage of destabilizing intramolecular hydrogen bonds containing hydroxyl group by water molecules.

Akin to polyserine, turns and coils in polyvaline are enriched by 3_10_-helix but only at *σ* = 0.776 nm^−2^ in a vacuum. When polyvaline interacts with CNT modified by the PEO chains at all grafting densities and with pristine CNT in water and saline, the coils and dominating turns constitute the secondary structure.

Since the 3_10_-helix represents a more compact arrangement of amino acid residues than α-helix, the occurrence of 3_10_-helices in investigated peptides instead of α-helices might be the consequence of restricted space experienced by the peptides near PEO-coated or pristine CNT.

### 3.3. Distance of Peptides from CNT

The distance of the center of mass of the peptides from the CNT sidewall, *D*, along with the radius of gyration of the peptides, *R*_g_, is summarized in Table 1 for CNT grafted with the PEO chains at different grafting densities and under different conditions. With a few exceptions, the distance of the center of mass from the CNT sidewall tends to increase with the increasing grafting density. Small deviations in the observed trends follow from the irregularities in the asymmetry of the peptides acquired during their adsorption on CNTs. Interesting is the closer contact of polyglycine with CNT in water or saline than in a vacuum. This finding is consistent with the density distributions and was rationalized in Section 3.1. Moreover, apart from the deeper insertion of polyglycine into the PEO layer in water or saline, an effort to maximize the favorable contacts between inserted polyglycine and CNT may lead to the elongation of polyglycine along the CNT axis, as in the case shown in Appendix A for *σ* = 0.388 nm^−2^, and to the concomitant reduction in size in the direction perpendicular to the CNT axis described below. In addition to the increased number of favorable interactions of polyglycine with CNT and the PEO chains, another factor that promotes the penetration of polyglycine into the PEO layer is the entropy gain due to water molecules expelled from the PEO layer. On the other hand, in a vacuum, the PEO chains as well as polyglycine are more compact, which prevents direct contact of polyglycine with the CNT sidewall. In line with the density distributions (Figure 2), the center of mass of polyvaline in a vacuum is found to be closer to the CNT sidewall than the center of mass of polyglycine and polyserine for the two lower grafting densities.

The center of mass of more sizeable polyserine and polyvaline does not penetrate through the grafted PEO layer in water or saline as readily. Appendix A collects the snapshots of equilibrated conformations of the PEO chains grafted on CNT at different grafting densities in a vacuum and water in the absence of the peptides. In a vacuum, CNT is compactly covered by the PEO chains at all grafting densities whereas in water, the chains are more sparsely distributed around CNT and take over brush conformations at *σ* = 0.776 nm^−2^ and *σ* = 1.553 nm^−2^. The conformations of PEO chains in saline (not shown) are essentially identical to those in water. The coarse-grained molecular dynamics simulation study reported more favorable adsorption of linear human fibrinogen than spherical human serum albumin onto CNT covalently functionalized or noncovalently covered with poly(ethylene glycol) [73]. The authors rationalized this difference in adsorption by the distinct shapes of the investigated proteins when a linear protein binds to an incompletely coated CNT more efficiently than a spherical protein. In the present study, the investigated peptides are much smaller in size than human fibrinogen or human serum albumin. In the case of peptides of smaller size and higher backbone flexibility, their shape may be modified to maximize contact with the CNT sidewall and PEO chains after penetration through the PEO layer. The loss of entropy caused by the restricted space of the adsorbed peptides may be enthalpically compensated through the favorable van der Waals and Coulomb interactions of the peptides with the CNT sidewall and the PEO chains.

The information on how the size and shape of the peptides are modified due to their adsorption on PEO modified or pristine CNT can be obtained from the radius of gyration, *R*_g_, and its component along the CNT axis, *R*_gz_, (Figure 6). With the exception of polyglycine adsorbed on pristine CNT in water and saline and on CNT covered by the PEO chains at *σ* = 0.388 nm^−2^ in water, where polyglycine undergoes asymmetry enhancement along the CNT axis, *R*_g_ and *R*_gz_ component are reduced after polyglycine adsorption when compared with polyglycine freely relaxed in water. In the remaining scrutinized systems, the overall radius of gyration of polyglycine decreases after its adsorption and drops below the *R*_g_ value of free polyglycine attained in water. In spite of this size reduction in a vacuum and in saline, polyglycine becomes more extended along the CNT axis at *σ* = 1.553 nm^−2^ as can be deduced from the reduced *R*_g_ value and unchanged *R*_gz_ value. On the other hand, at this grafting density in water, the polyglycine size is reduced in symmetrical way. This implies that the shape of polyglycine after its adsorption on CNT is controlled by its initial arrangement with respect to the CNT axis rather than by the environment. As reflected on the increased van der Waals and Coulomb contributions to the binding free energy discussed below, these elongated shapes of polyglycine allow more contacts between polyglycine and the CNT sidewall or PEO chains.

Polyserine possesses conformation elongated along the axis of bare CNT in water. Consistently with the aforementioned close approach of polyserine to CNT grafted with the PEO chains at *σ* = 0.388 nm^−2^ in saline, the marked elongation of the shape of polyserine along the CNT axis may follow from the maximization of favorable interactions of polyserin with CNT or the PEO chains including also sodium bridges between oxygen atoms of polyserine and oxygen atoms of the PEO chains.

In all investigated media, the size of polyvaline is slightly reduced during its adsorption on the PEO coated as well as on pristine CNT. Since this reduction occurs also in the longitudinal direction, it can be assumed that the adsorption of polyvaline is not accompanied by an enhancement of shape asymmetry. Among the investigated peptides, the radius of gyration of polyvaline is the least affected by the PEO coverage of CNT and the nature of environment. The standard deviations of *R*_g_ and *R*_gz_ span the range 1–5% of the average values in vacuum and 3–12% in water or saline and the standard deviations of *D* span the range 8–12% of the average values.

The dynamics of conformational changes of the peptides can be monitored using the time evolution of the root-mean-square deviation, RMSD, of atom positions with respect to the initial conformation, which is evaluated as follows:(1)$$\mathrm{RMSD}\left(t\right)={\left(\frac{1}{N}\overset{N}{\underset{i=1}{\sum }}||r_{i}\left(t\right)-r_{i}{\left(0\right)||}^{2}\right)}^{\frac{1}{2}}$$
where *N* it the number of atoms ***r***_i_(*t*) is the position of atom *i* at time *t* of the structure superimposed with the reference structure at time *t* = 0. In a vacuum, this quantity levels off readily, mainly for polyglycine, and its fluctuations are damped with respect to its fluctuations in water and saline. While during the initial phase of MD, i.e., the equilibration period, RMSD reaches the plateau in the first 100 ps in a vacuum, in water and saline, RMSD saturates after 4–6 ns (Appendix A).

### 3.4. Binding Free Energy

The biding free energy of adsorption of the peptide on PEO modified or pristine CNT, Δ*G*_bind_, in a solvent is the free energy difference between the composed system, $G_{S}^{\mathrm{PC}}$, and the sum of the free energy of the individual components, i.e., the peptide, $G_{S}^{P}$, and CNT, $G_{S}^{C}$
(2)$$\Delta G_{\mathrm{bind}}=G_{S}^{\mathrm{PC}}-\left(G_{S}^{P}+G_{S}^{C}\right)$$

The binding free energy was calculated using the molecular mechanics Poisson-Boltzmann surface area (MM-PBSA) method [93,94,95]. The calculation of the binding free energy is derived from the thermodynamic cycle


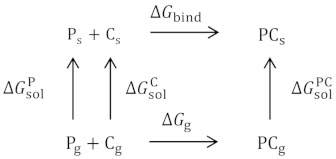
(3)
where $\Delta G_{\mathrm{sol}}^{P}$, $\Delta G_{\mathrm{sol}}^{C}$, and $\Delta G_{\mathrm{sol}}^{\mathrm{PC}}$ are the solvation free energies of the peptide, CNT, and their complex, respectively, and $\Delta G_{g}$ is the binding free energy in the gaseous state. The binding free energy can be expressed with respect to the binding free energy in gaseous state as follows:(4)$$\Delta G_{\mathrm{bind}}=\Delta G_{g}+\Delta G_{\mathrm{sol}}^{\mathrm{PC}}-\Delta G_{\mathrm{sol}}^{P}-\Delta G_{\mathrm{sol}}^{C}$$

The binding free energy for the system in the gaseous state can be decomposed as follows:(5)$$\Delta G_{g}=\left(\Delta \langle {\;E}_{\mathrm{intra}}\rangle +\Delta \langle E_{\mathrm{vdW}}\rangle +\Delta \langle E_{\mathrm{coul}}\rangle \right)-T\Delta \langle S_{\mathrm{MM}}\rangle$$
where *E*_intra_, *E*_vdW_, and *E*_coul_ stand for the potential energy due to all intramolecular, van der Waals, and Coulomb interactions, respectively, and *T*Δ*S*_MM_ is the entropic contribution. Estimating the change in the internal entropy Δ*S*_MM_ would require an additional computationally expensive step in the calculations which can be performed using normal-mode analysis method or methods based on empirical conjecture of the average entropic cost of constraining rotation around a given torsional degree of freedom. The *T*Δ*S*_MM_ term is usually much smaller than the remaining terms, and it has been shown that this term does not improve the prediction accuracy of the binding free energy [96]. Moreover, when evaluating the binding free energy trend rather than the absolute values of the binding free energy, omission of this contribution appears to be a reasonable approximation. The average over the generated conformations is denoted by angle brackets. The solvation free energies consist of the polar and nonpolar contributions, i.e.:(6)$$\Delta G_{\mathrm{sol}}^{X}=\Delta G_{\mathrm{pol}}^{X}+\Delta G_{\mathrm{nonpol}}^{X}$$

The polar term $\Delta G_{\mathrm{pol}}^{X}$ is the energy required to transfer the component X from the gas phase to the polar solvent and is obtained by solving the Poisson-Boltzmann equation. The nonpolar term $\Delta G_{\mathrm{nonpol}}^{X}$ consists of the contribution due to formation of a cavity in the solvent and the van der Waals energy due to the attractive interactions between solvent and solute. In this study, the nonpolar term only accounts for the energy of cavity formation and is estimated from the solvent accessible surface area as the surface free energy. The reported values of the binding free energies are the averages over the frames generated during the last 10 ns of the simulations.

The calculated binding free energies of the composed system built up of PEO coated or pristine CNT and the three investigated peptides in a vacuum, water, and saline are presented in Figure 7. Negative van der Waals energy changes caused by the peptide adsorption represent the dominant contribution to the binding free energy in all simulated systems (Appendix A). For the systems in a vacuum, the van der Waals contribution to the free energy exceeds the van der Waals contribution for the corresponding systems in water and saline. This is because more interaction sites on the CNT sidewall and in the PEO chains are available for interactions with the peptides in a vacuum. In saline and water, many interaction sites of the CNT sidewall and the PEO chains are occupied by water molecules. Upon insertion of the peptide into the grafted PEO layer, some interactions of PEO or CNT atoms with water molecules are replaced by the interactions with the peptide and some interaction sites remain inaccessible to the peptide. Both of these factors reduce the favorable van der Waals interactions in comparison to the situation in a vacuum. On the other hand, the released water molecules, contribute positively to the entropy of the system. In agreement with the cylindrical density distribution functions (Figure 2) and the elongated conformations (Figure 6), the exceptions are found and explained above for polyglycine in water and for polyserine in saline adsorbed on CNT grafted with the PEO chains at σ = 0.388 nm^−2^. In these systems, the deeper insertion of polyglycine or polyserine into the PEO layer and their elongated shape impart the opportunity to increase the number of favorable van der Waals interactions of the peptides with the CNT sidewall or the PEO chains. The interactions between polyvaline and the PEO chains grafted on CNT in water favorably affect the van der Waals attraction as can be deduced from the more negative values of the van der Waals contribution in the case of CNT covered with the PEO chains at σ = 0.388 nm^−2^ and 0.776 nm^−2^ when compared with the van der Waals contribution for pristine CNT in water although polyvaline does not penetrate to the PEO modified CNT sidewall as closely as to pristine CNT (Figure 2). Although less pronounced, such van der Waals attraction between the peptide and the PEO chains can also be observed for polyvaline and polyserine in saline.

The negative Coulomb contribution to the binding free energy (Appendix A) is also more significant in a vacuum than in water and saline. However, this contribution is less significant than the van der Waals contribution in all three media because only the PEO chains exhibit the Coulomb interactions with the adsorbed peptides. Moreover, the electrostatic interactions between the peptides and the PEO chains in aqueous environment may be shielded by water molecules. As expected, the smallest Coulomb contribution among the investigated peptides is found for hydrophobic polyvaline. Although the PEO coverage diminishes the van der Waals interactions between the peptides and the CNT sidewall, this reduction is compensated by the creation of new Coulomb and van der Waals interaction contacts between the peptides and the PEO chains. It should be recalled at this point that CNT does not exhibit electrostatic interactions in this simulation model; hence, the Coulomb contributions to the binding free energy in systems with pristine CNT might be underestimated.

The contribution to the binding free energy due to the polar solvation energy (Appendix A) is always positive, more significantly for PEO covered CNT than for pristine CNT. Similar trend applies to the negative contribution of the nonpolar solvation energy (Appendix A) which is in absolute values an order of magnitude lower than the contribution of the polar solvation energy.

The overall binding free energy is higher (in absolute values) in vacuum than in water and saline. The grafting density does not define any trend in the binding free energy values since the diminished van der Waals interactions between the peptides and CNT caused by the presence of the PEO chains may be replaced by the van der Waals and Coulomb interactions between the peptides and the PEO chains. Moreover, the complexity is accentuated by the presence of water molecules, which at the one hand occupy some interaction sites at the PEO covered CNT, and on the other hand, their expulsion from the PEO layer contributes to an increase in the entropy of the system. The highest binding free energy (in negative values) in systems with pristine CNT is found for polyglycine (−316.04 kJ·mol^−1^ in water and −290.21 kJ·mol^−1^ in saline) followed by polyvaline (−244.84 kJ·mol^−1^ in water and −244.27 kJ·mol^−1^ in saline) and polyserine (−231.41 kJ·mol^−1^ in water and −205.36 kJ·mol^−1^ in saline). In vacuum, polyserine turns out to be adsorbed most strongly on the PEO-coated CNT.

The contributions due to the peptide-CNT and peptide-PEO interactions depends on the insertion of the peptide into the PEO layer, which in turn is influenced by the size and flexibility of the peptide as well as its initial conformation and orientation with respect to the CNT axis. In the investigated systems, coating of CNT with the PEO chains does not necessarily reduce the binding free energy as is evident for polyvaline interacting with CNT grafted with the PEO chains at σ = 0.388 nm^−2^ and = 0.776 nm^−2^ in water and saline or polyserine interacting with CNT grafted with the PEO chains at σ = 0.388 nm^−2^ in saline. In the former case, this finding is caused by the van der Waals interactions between polyvaline and PEO chains, whereas in the latter case, the observed trend is dictated by the deeper insertion of polysaline in the PEO layer.

## 4. Conclusions

Interactions of polyglycine, polyserine, and polyvaline with pristine CNT and CNT irreversibly grafted with the PEO chains at different grafting densities in vacuum, water, and saline have been studied using atomistic MD simulations. Peptide adsorption is not prevented by covering the CNT sidewalls with PEO chains, hydration of the PEO layer, or the presence of NaCl. According to the cylindrical density distribution of the PEO chains, the peptides adhere to the grafted PEO chains and penetrate through the PEO layer without compressing the PEO layer. The influence of the aqueous environment on the adsorption of the peptides is manifested at lower grafting densities. Water molecules disrupt the hydrogen bonds formed between the peptides and the PEO chains. Peptide adsorption is insensitive to the presence of NaCl apart from the special case of polyserine in saline at σ = 0.388 nm^−2^ when the sodium bridges between the oxygen atoms of polyserine and the oxygen atoms of the PEO chains are formed. The size and backbone flexibility of the investigated peptides determine the efficiency of peptides insertion into the PEO layer by maximizing favorable contacts with the PEO and CNT interaction sites. The entropy loss due to the restricted space experienced by the peptides upon their adsorption is compensated entropically by released water molecules from the PEO layer and enthalpically by maximizing the favorable van der Waals and Coulomb interactions through conformational changes in the peptides.

Adsorption of the investigated peptides on the PEO coated CNT in aqueous environment is a complex phenomenon governed by many factors such as water affinity, flexibility, size, shape, and secondary structure of the peptides, as well as conformation and grafting density of the grafted PEO chains, their cylindrical density distribution, and hydration. The number of interaction sites available on the CNT sidewall for the peptides during their adsorption decreases with increasing grafting density of the PEO chains; at the same time, the number of potentially new interaction sites in the layer of the PEO chains available for the interaction with the peptides increases. The availability of the interaction sites in the PEO layer descends with the increasing level of hydration of the PEO layer, i.e., with decreasing grafting density of the PEO chains. However, higher number of water molecules can be expelled from the more hydrated PEO layer upon the peptide adsorption, which entropically drives the peptide adsorption (hydrophobic effect). These opposed effects make the situation more complex. Thus, it appears that instead of defining some trend in the peptide adsorption with respect to the scrutinized parameters, rather, optimal conditions for protein repulsion or suppressed protein adsorption should be sought.

## Figures and Tables

**Figure 1 nanomaterials-12-03795-f001:**
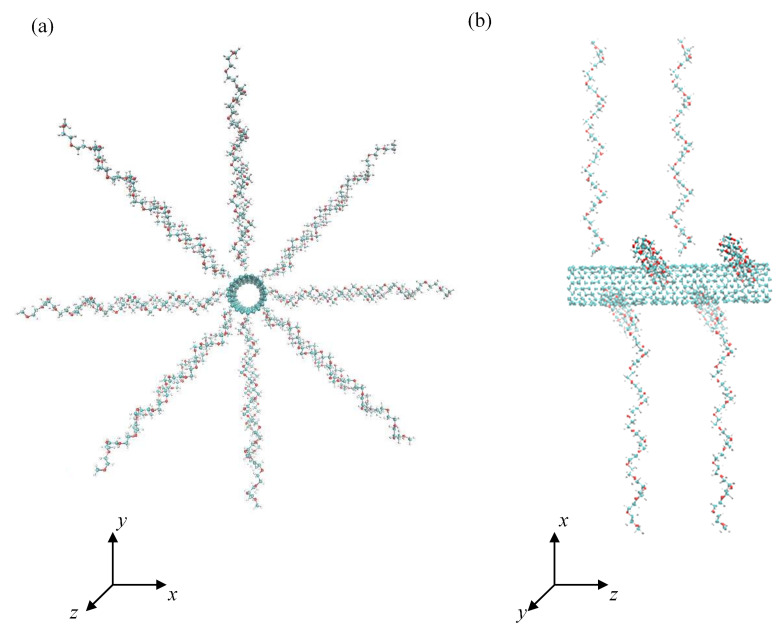
Front (**a**) and side view (**b**) of the initial conformation of CNT grafted with 8 PEO chains at grafting density 0.776 nm^−2^.

**Figure 2 nanomaterials-12-03795-f002:**
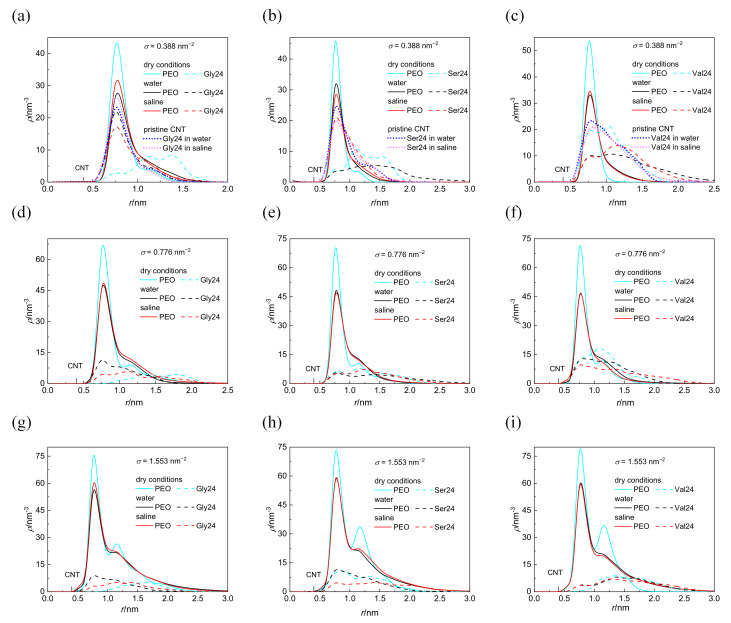
Cylindrical density distribution of the PEO chains grafted at *σ* = 0.388 nm^−2^ (**a**–**c**), 0.776 nm^−2^ (**d**–**f**), and 1.553 nm^−2^ (**g**–**i**) and polyglycine (**a**,**d**,**g**), polyserine (**b**,**e**,**h**), and polyvaline (**c**,**f**,**i**). For comparison, the cylindrical density distributions of peptides interacting with pristine CNT in water and saline are also presented (**a**–**c**). The position of the CNT sidewall is also marked.

**Figure 3 nanomaterials-12-03795-f003:**
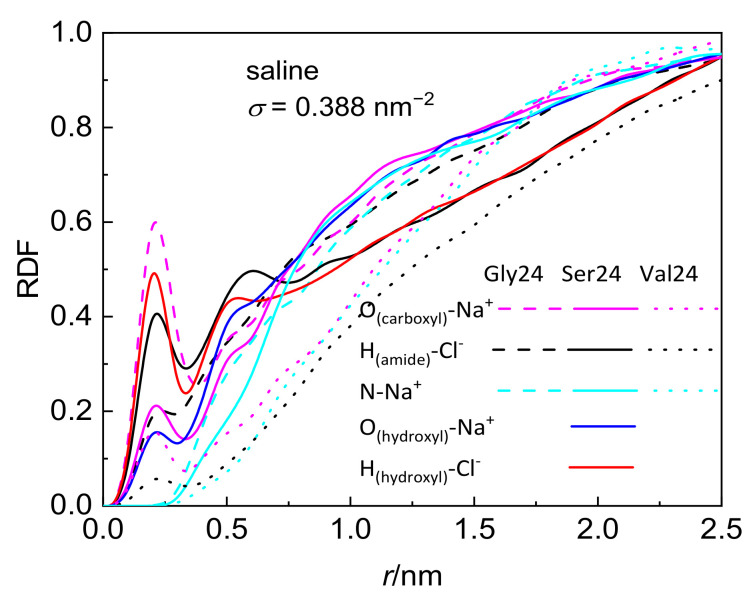
Radial distribution function of pairs comprising selected peptide atoms and sodium cations or chloride anions in systems composed of the peptides interacting with CNT grafted with the PEO chains at *σ* = 0.388 nm^−2^.

**Figure 4 nanomaterials-12-03795-f004:**
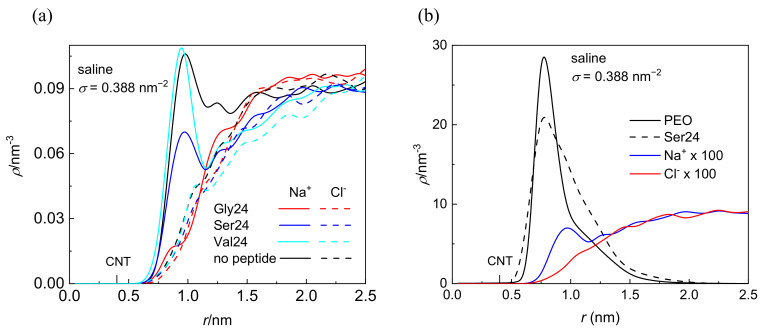
Cylindrical density distribution of sodium cations and chloride anions in systems containing CNT grafted with the PEO chains at *σ* = 0.388 nm^−2^ in the presence or absence of the peptides (**a**) and of heavy atoms of the PEO chains, all atoms of polyserine, sodium cations, and chloride anions in a system composed of polyserine and CNT grafted with the PEO chains at *σ* = 0.388 nm^−2^; the values of ions are rescaled by the numerical factor of 100 (**b**). The position of the CNT sidewall is also marked.

**Figure 5 nanomaterials-12-03795-f005:**
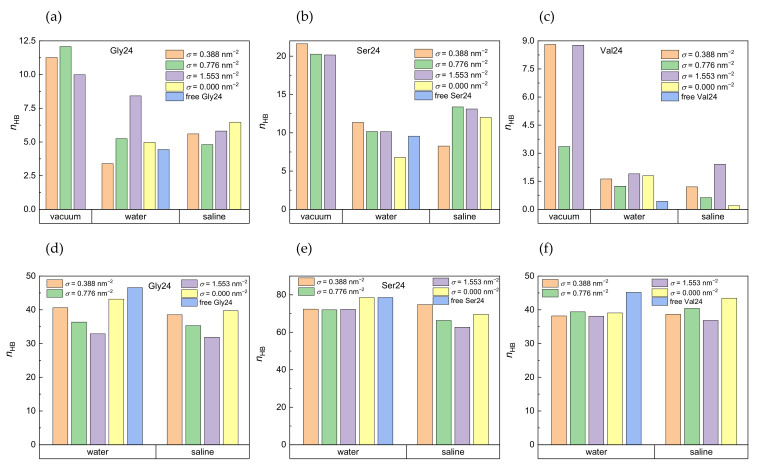
Number of intramolecular hydrogen bonds in polyglycine (**a**), polyserine (**b**), and polyvaline (**c**) in a vacuum, water, and saline and number of hydrogen bonds formed by water molecules with polyglycine (**d**), polyserine (**e**), and polyvaline (**f**) in water and saline in systems with different grafting densities. For comparison, the number of hydrogen bonds for the free peptides and the peptides adsorbed on pristine CNT are also included. Standard deviations are in the range of 5–12% of the average values.

**Figure 6 nanomaterials-12-03795-f006:**
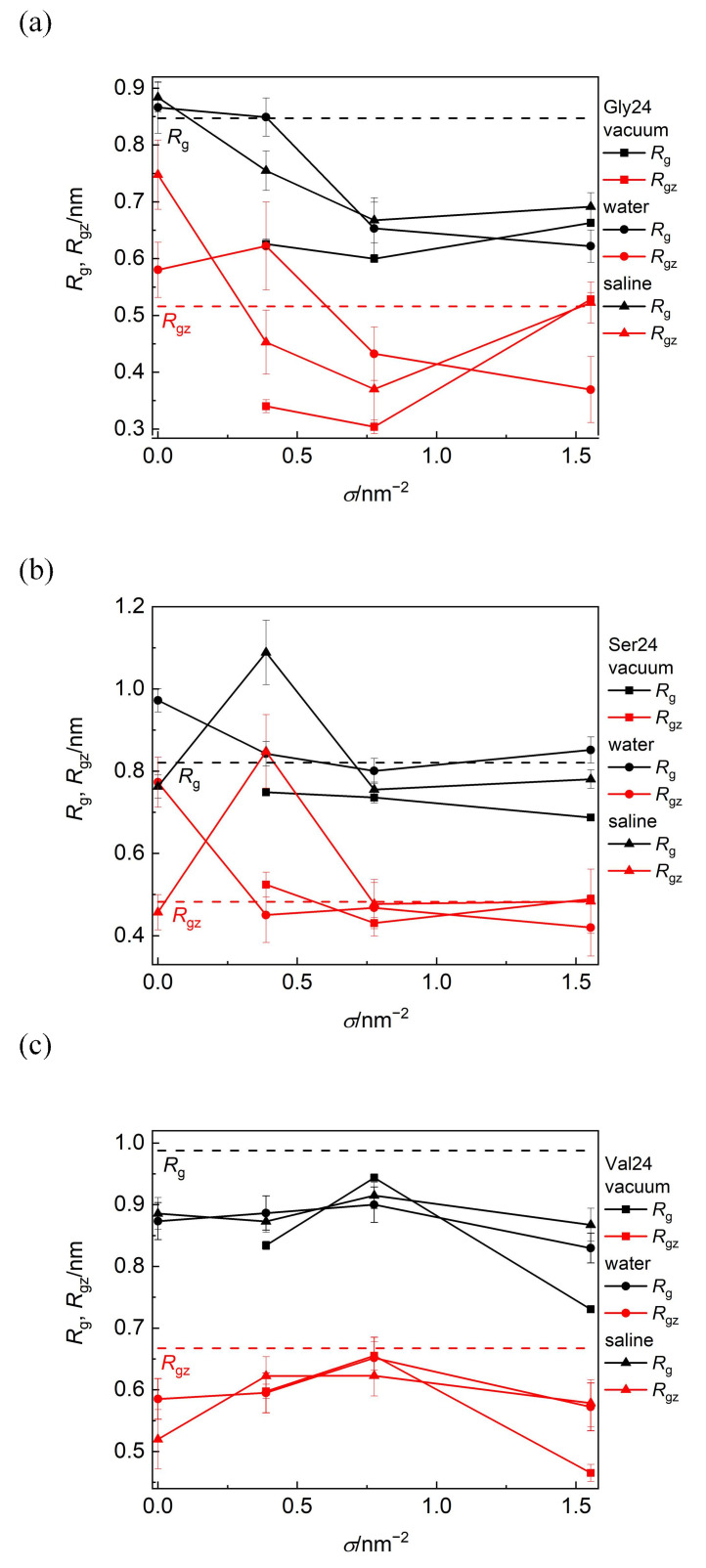
Radius of gyration, *R*_g_, and its component along the CNT axis, *R*_gz_, of polyglycine (**a**), polyserine (**b**), and polyvaline (**c**) as functions of grafting density in a vacuum, water, and saline. The black and red dashed lines represent the values of *R*_g_ and *R*_gz_ for free peptides in water, respectively.

**Figure 7 nanomaterials-12-03795-f007:**
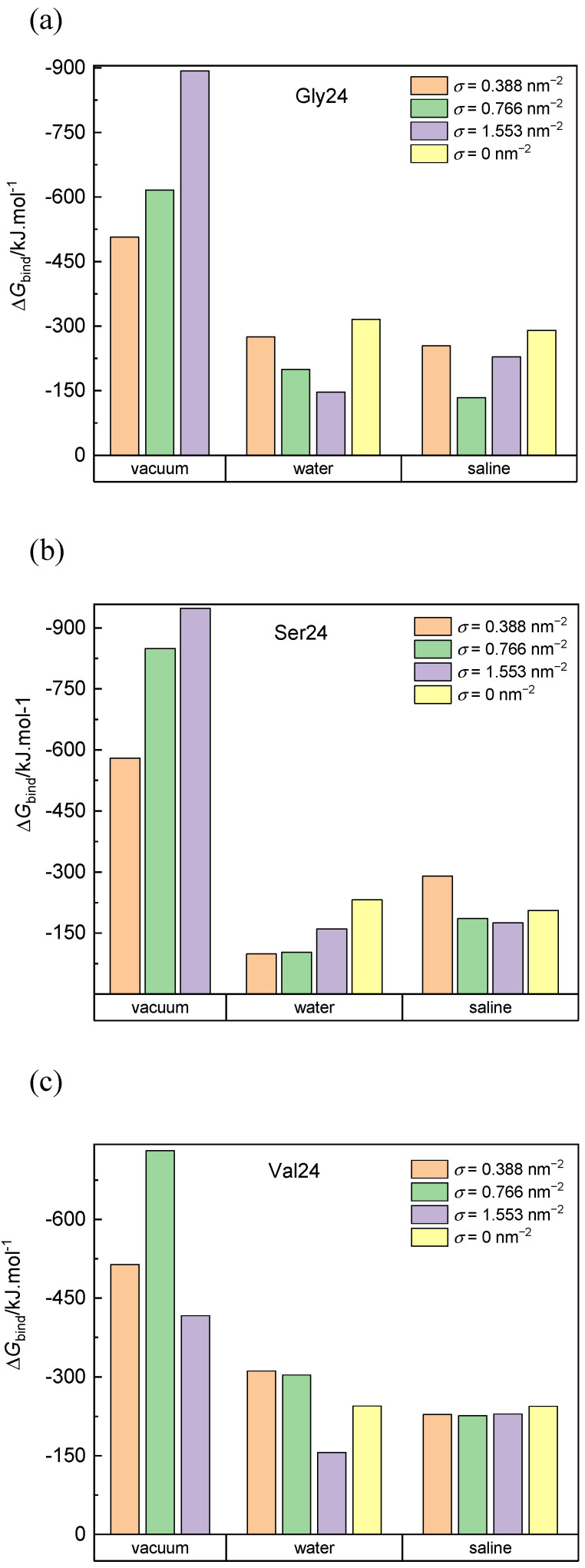
Binding free energy of polyglycine (**a**), polyserine (**b**), and polyvaline (**c**) interacting with pristine CNT and CNT grafted with the PEO chains at different grafting densities in a vacuum, water, and saline. Standard deviations are approximately 5% and 10% of the average values for systems in vacuum and aqueous environment, respectively.

**Table 1 nanomaterials-12-03795-t001:** Distance of the center of mass of the peptides from the CNT sidewall, *D*, grafted with the PEO chains at different grafting densities, *σ*, and the radius of gyration, *R*_g_, in a vacuum, water, and saline.

Polyglycine			Polyserine		Polyvaline	
Vacuum						
*σ*/nm^−2^	*D*/nm	*R*_g_/nm	*D*/nm	*R*_g_/nm	*D*/nm	*R*_g_/nm
0.388	0.763	0.626	0.848	0.749	0.555	0.834
0.776	1.258	0.600	0.943	0.736	0.614	0.944
1.553	1.250	0.663	0.784	0.687	1.256	0.731
Water *						
*σ*/nm^−2^	*D*/nm	*R*_g_/nm	*D*/nm	*R*_g_/nm	*D*/nm	*R*_g_/nm
0.000	0.191	0.866	0.458	0.972	0.467	0.873
0.388	0.339	0.849	1.169	0.842	0.913	0.886
0.776	0.581	0.653	1.226	0.801	0.808	0.900
1.553	0.700	0.622	0.696	0.852	1.259	0.830
Saline						
*σ*/nm^−2^	*D*/nm	*R*_g_/nm	*D*/nm	*R*_g_/nm	*D*/nm	*R*_g_/nm
0.000	0.324	0.884	0.493	0.763	0.538	0.886
0.388	0.389	0.755	0.409	1.088	0.804	0.873
0.776	0.926	0.667	1.019	0.755	1.080	0.915
1.553	0.978	0.691	1.253	0.780	1.314	0.867

* The radius of gyration for free polyglycine, polyserine, and polyvaline in water after 10 ns of MD simulations are 0.847 nm, 0.821 nm, and 0.988 nm, respectively.

## Data Availability

Data are available in the manuscript, as Supporting Materials. and on request from the corresponding author.

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
