# Peer review of "Adsorption of Peptides onto Carbon Nanotubes Grafted with Poly(ethylene Oxide) Chains: A Molecular Dynamics Simulation Study"

_nanomaterials, 2022, doi:10.3390/nano12213795_

Round 1

Reviewer 1 Report

Comments from Reviewer

Title: Adsorption of peptides onto carbon nanotubes grafted with poly(ethylene oxide) chains: A Molecular dynamics simulation study

The current form's presentation of methods and scientific results is satisfactory for publication in the Nanomaterials journal. The minor and significant drawbacks to be addressed can be specified as follows:
1.    I suggest writing an abstract with descriptive and concise information about this research. A standard abstract definition is: "An abstract is a concise summary of an experiment or research project. It should be brief -- typically under 200 words. The purpose of the abstract is to summarize the research paper by stating the purpose of the research, the experimental method, the findings, and the conclusions."
2.    Line 192. Periodic conditions? Simulation box size?
3.    How many water molecules were there in the simulation box?
4.    How many times were simulations repeated for a given system?
5.    Fig. 2. Why are there different ranges of the x-axis?
6.    Fig. 2 (a)-(c). These panels are unreadable. Dashed lines repeated twice — similar line colours.
7.    Fig. 2(h). 75 is cut off.
8.    Fig. 2(h), black solid line. Molecules inside CNT? Please, look at the black line near the bar related to CNT.
9.    Fig. 6. It is complicated to analyze this figure without error bars. Especially if there are visible deviations from the trend in the curves.
10.    The conclusion presented in this manuscript is not in the format of a conclusion for a scientific article. According to the literature: "The Conclusion section presents the outcome of the work by interpreting the findings at a higher level of abstraction than the Discussion and by relating these findings to the motivation stated in the Introduction." They are dramatically long. Please shorten them.

Sincerely,
    The reviewer.

Author Response

Reviewer 1

1. I suggest writing an abstract with descriptive and concise information about this research. A standard abstract definition is: "An abstract is a concise summary of an experiment or research project. It should be brief -- typically under 200 words. The purpose of the abstract is to summarize the research paper by stating the purpose of the research, the experimental method, the findings, and the conclusions."

Response: We have reduced the text in the abstract according to this suggestion.

2. Line 192. Periodic conditions? Simulation box size?

Response: We have added the following sentences

Periodic boundary conditions were applied in all three dimensions. The size of the cubic simulation box was 8-9 nm in peptide-free systems and 11-12 nm in peptide-containing systems. In the case of pristine CNTs, the size of the cubic simulation box was 6-6.5 nm.

3. How many water molecules were there in the simulation box?

Response: We have added the following sentence

The number of water molecules varied depending on the simulation box size from 7 010 (simulation box size 6 nm) to 55 966 (simulation box size 12 nm).

4. How many times were simulations repeated for a given system?

Response: We tested 3 simulation runs for some selected systems and found out that the resulting studied quantities did not change when compared with a single run. Therefore, we considered one simulation run for each system. The following sentence has been added

One simulation run was performed for each system.

5. Fig. 2. Why are there different ranges of the x-axis?

Response: The range of x-axis was set to cover the density distribution function. We did not consider the distances at which the density distribution functions were vanished.

6. Fig. 2 (a)-(c). These panels are unreadable. Dashed lines repeated twice — similar line colours.

Response: We have changed the line types to short dashed lines and the green color to light magenta.

7. Fig. 2(h). 75 is cut off

Response: We have removed this drawback.

8. Fig. 2(h), black solid line. Molecules inside CNT? Please, look at the black line near the bar related to CNT.

Response: We thank the reviewer for this notice. We mistakenly included a plot corresponding to a system where one PEO chain was slightly pushed behind the edge of CNT and bent toward the CNT axis. We did consider such phenomena and run new simulations in such a case. We have now replaced the plot by the correct one.

9. Fig. 6. It is complicated to analyze this figure without error bars. Especially if there are visible deviations from the trend in the curves.

Response: We have added the error bars to the plots in Figure 6.

10. The conclusion presented in this manuscript is not in the format of a conclusion for a scientific article. According to the literature: "The Conclusion section presents the outcome of the work by interpreting the findings at a higher level of abstraction than the Discussion and by relating these findings to the motivation stated in the Introduction." They are dramatically long. Please shorten them.

Response: We have shortened the Conclusions.

Reviewer 2 Report

The current study by Benková and co-authors reported the adsorption of peptides onto carbon nanotubes grafted with poly(ethylene oxide) chains by performing molecular dynamics simulations. The peptides with different hydropathies have been considered for studying the interactions when binding with CNT with and without PEO coating (and of course in three different mediums). Density distributions, number of hydrogen bonds, interacting distances and binding free energies are reported and analyzed in detail. I think the work should be published, but before that, the authors should consider addressing the following issues:

Majors:

1.     I appreciate that the authors have covered all the relevant literature in the field, but I think the introduction is a little bit tedious, the authors should consider simplifying the introduction and emphasizing the unsolved questions, and to make it more attractive to the potential readers.

2.     For Line 203, the authors wrote: “The hydropathy indices of valine, glycine, and serine are 4.2, −0.4, and −0.8, respectively”, what was the reason for the authors to choose the components with similar hydropathies (for the second and third ones)?

3.     For equation 5, the authors should explain more about the calculation of the entropic contribution TΔSMM.

4.     The conclusion part is also tedious, the authors should only focus on the most pronounced outcomes.

Minors:

i.                For Line 87, primary, secondary, and ternary adsorptions need to be briefly introduced.

ii.              For Line 195, it is better to mention the total area of the CNT.

Author Response

Reviewer 2

1. I appreciate that the authors have covered all the relevant literature in the field, but I think the introduction is a little bit tedious, the authors should consider simplifying the introduction and emphasizing the unsolved questions, and to make it more attractive to the potential readers.

Response: We provide an overview of fundamental studies performed for planar PEO coated surfaces where the theory is more thoroughly developed compared with the curved counterparts. This is to familiarize the reader with existing theories explaining the conformational behavior of the PEO chains grafted onto planar substrates which is crucial for the interactions with proteins and peptides. We have now shortened the section dealing with planar surfaces by omitting the conclusions of the cited studies, and we have also reduced the last paragraph where the information presented in the Methodology and Results and discussion subsections is provided.

2. For Line 203, the authors wrote: “The hydropathy indices of valine, glycine, and serine are 4.2, −0.4, and −0.8, respectively”, what was the reason for the authors to choose the components with similar hydropathies (for the second and third ones)?

Response: Although serine has a similar hydropathy index compared to glycine, we were interested in the effects arising from the presence of the hydroxyl group responsible for specific interactions with water and the PEO chains through hydrogen bonding. We have added the following sentence after the mentioned sentence

Although serine has a similar hydropathy index compared to glycine, serine contains a hydroxyl group that is responsible for specific interactions with water and the PEO chains.

3. For equation 5, the authors should explain more about the calculation of the entropic contribution TΔSMM.

Response: We have rephrased this part. In fact, we have not considered this entropy contribution in the binding free energy as we have rationalized in the rephrased text as follows

Estimating the change in the internal entropy ΔSMM would require an additional computationally expensive step in the calculations which can be performed using normal-mode analysis method or methods based on empirical conjecture of the average entropic cost of constraining rotation around a given torsional degree of freedom. The TΔSMM term is usually much smaller than the remaining terms, and it has been shown that this term does not improve the prediction accuracy of the binding free energy.96 Moreover, when evaluating the binding free energy trend rather than the absolute values of the binding free energy, omission of this contribution appears to be a reasonable approximation.

We have included the corresponding reference in the References section.

(96)    Hou, T.; Wang, J.; Li, Y.; Wang, W. Assessing the Performance of the MM/PBSA and MM/GBSA Methods . 1 . The Accuracy of Binding Free Energy Calculations Based on Molecular Dynamics Simulations. J. Chem. Inf. Model. 2011, 69–82.

4. The conclusion part is also tedious, the authors should only focus on the most pronounced outcomes.

Response: We have shortened the Conclusions.

Minors

For Line 87, primary, secondary, and ternary adsorptions need to be briefly introduced.

Response: We have added the following sentence

Primary adsorption, typical of small proteins, occurs at the surfaces, secondary adsorption, characteristic of large rod-like proteins, occurs at the edge of the grafted brushes, and ternary adsorption is observed when a globular protein penetrates the brush.

For Line 195, it is better to mention the total area of the CNT.

Response: We have added the following sentence

The total area of the CNT sidewall was 10.3 nm2.

Round 2

Reviewer 1 Report

I believe that there is surely an improvement in the quality of the manuscript. This work is likely to be interesting to the readers of the journal. It may now be considered for possible publication.